# Estimating the risk of species interaction loss in mutualistic communities

**Benno I. Simmons**[1,2]*, **Hannah S. Wauchope**[1], **Tatsuya Amano**[1,3,4], **Lynn V. Dicks**[5,6], **William J. Sutherland**[1], **Vasilis Dakos**[7]

**1** Conservation Science Group, Department of Zoology, University of Cambridge, Cambridge, United Kingdom, **2** Centre for Ecology and Conservation, College of Life and Environmental Sciences, University of Exeter, Penryn, United Kingdom, **3** Centre for the Study of Existential Risk, University of Cambridge, Cambridge, United Kingdom, **4** School of Biological Sciences, University of Queensland, Brisbane, Australia, **5** School of Biological Sciences, University of East Anglia, Norwich, United Kingdom, **6** Agroecology Group, Department of Zoology, University of Cambridge, Cambridge, United Kingdom, **7** Institut des Sciences de l'Evolution (ISEM), CNRS, Univ Montpellier, EPHE, IRD, Montpellier, France

* benno.simmons@gmail.com

**Data Availability Statement:** All networks are publicly available from the Web of Life repository (www.web-of-life.es) and Data Dryad (https://doi.org/10.5061/dryad.76173). The raw data for S5

## Abstract

Interactions between species generate the functions on which ecosystems and humans depend. However, we lack an understanding of the risk that interaction loss poses to ecological communities. Here, we quantify the risk of interaction loss for 4,330 species interactions from 41 empirical pollination and seed dispersal networks across 6 continents. We estimate risk as a function of interaction vulnerability to extinction (likelihood of loss) and contribution to network feasibility, a measure of how much an interaction helps a community tolerate environmental perturbations. Remarkably, we find that more vulnerable interactions have higher contributions to network feasibility. Furthermore, interactions tend to have more similar vulnerability and contribution to feasibility across networks than expected by chance, suggesting that vulnerability and feasibility contribution may be intrinsic properties of interactions, rather than only a function of ecological context. These results may provide a starting point for prioritising interactions for conservation in species interaction networks in the future.

## Introduction

Species are the predominant biological unit of interest across ecology and conservation. However, it is interactions between species, rather than species themselves, that mediate the ecological functions that drive community dynamics and support biodiversity [1]. For example, pollination interactions shape co-evolution in diverse plant–animal communities [2], the structure of ecological networks shapes the persistence of mutualistic systems [3,4], and seed dispersal maintains spatial patterns of diversity [5]. Given the importance of interactions—both with other species and with the abiotic environment—for ecosystem functioning and stability, their loss could have reverberating effects on entire communities and, ultimately, the ecosystem services they deliver [6,7].

Analysis can be found at https://doi.org/10.5061/dryad.dncjsxkw2. The data underlying all plots except Figure A in S3 Analysis can be found at https://doi.org/10.6084/m9.figshare.12689258.v1. The data underlying Figure A in S3 Analysis can be found at https://doi.org/10.5061/dryad.cr3ft.

**Funding:** BIS is supported by the Natural Environment Research Council (https://nerc.ukri.org) as part of the Cambridge Earth System Science NERC DTP [NE/L002507/1] and a Royal Commission for the Exhibition of 1851 Research Fellowship (https://www.royalcommission1851.org). HSW was supported by a Cambridge Trust Cambridge-Australia Scholarship and a Cambridge Department of Zoology JS Gardiner Fellowship. TA was supported by the Grantham Foundation for the Protection of the Environment, the Kenneth Miller Trust and the Australian Research Council Future Fellowship (FT180100354). LVD was supported by the Natural Environment Research Council (https://nerc.ukri.org) (grants NE/K015419/1 and NE/N014472/1). WJS is funded by Arcadia. The funders had no role in study design, data collection and analysis, decision to publish, or preparation of the manuscript.

**Competing interests:** The authors have declared that no competing interests exist.

Although interactions are a vital component of biodiversity, they remain largely neglected in the presence of the dominant species-centred perspective [8,9]. The few existing studies of interaction loss tend to focus on the impact of anthropogenic stressors on single interactions at single sites [10]. Other studies that have considered interaction loss at the community level either are at local scales [11], are based on hypothetical network structures [12], or do not consider population dynamics [13]. Theory has shown that the reorganisation and changes in the number and structure of interactions can affect the stability and persistence of species in food webs [14,15], as well as mutualistic [16–18] and host–parasite communities [19,20]. However, this work has focused almost entirely on the overall structure of interactions within a community (metrics such as connectance and nestedness) and rarely on individual interactions themselves. There is thus a gap in assessing community-level responses to the loss of individual interactions. Specifically, we lack a quantitative understanding of the risk that interaction loss poses to the stability and persistence of communities.

Here, we address this gap by quantifying the risk of interaction loss to 41 pollination and seed dispersal communities that, combined, compose a global dataset of 4,330 species–species links (see Methods). Such mutualisms are fundamental to the functioning of most communities. The loss of pollination can lead to pollen limitation, potentially compromising reproduction for the vast majority of plant species that rely, to some extent, on animal pollinators [21,22]. Similarly, the disruption of seed dispersal can have deleterious, cascading consequences for those woody plant species that depend on frugivores, which can exceed 90% in biodiverse ecosystems such as tropical rainforests [23]. Loss of these mutualistic links can occur for many reasons, perhaps most obviously if one of the interacting species goes extinct. However, there are many other ways that a link can disappear without extinctions—such as phenological decoupling, changing behaviour, and ecological extinction (when a species is reduced to such low abundance that it no longer significantly interacts with other species) [24–27]—and it is this kind of link loss that is the focus of this study. Hereafter we distinguish between the terms "interaction" and "link": interaction refers to all occurrences of a given taxon–taxon interaction identity at the metaweb level (all local networks considered), whereas link refers to a single occurrence of an interaction in a particular local network. For example, let there be two networks, 1 and 2, with links between species X and species Y in a network represented as X–Y. Network 1 contains three links (A–B, A–C, and B–D), and Network 2 contains two links (A–B and A–E). Across these networks, there are five links (A–B, A–C, B–D, A–B, and A–E), but only four interactions (A–B, A–C, B–D, and A–E), because the link between species A and B occurs in two networks.

Conventionally, risk is a function of both the likelihood of an event occurring and the severity of the impacts if it does occur [28]. For ecological networks, we therefore start by reasoning that the risk of losing a particular link is a function of (i) the likelihood of that link being lost (that we refer to as link vulnerability, $V$) and (ii) the severity of the consequences to the community if the link is lost (that we measure as link contribution to network feasibility, $I$, a measure that reflects the extent to which a link contributes to a community's ability to tolerate environmental perturbations). In some ways, our 2 components of risk reflect the approach taken in systematic conservation planning, in which sites are prioritised for designation as protected areas based on their vulnerability and irreplaceability [29]. Using novel quantitative methods that we illustrate in Fig 1, we explore the relationship between vulnerability and feasibility contribution of all links in our dataset. We next examine whether vulnerability and feasibility contribution are intrinsic attributes of interactions, rather than functions of ecological context, by testing whether an interaction's vulnerability and feasibility contribution is more similar across occurrences than expected by chance. These relationships will help us

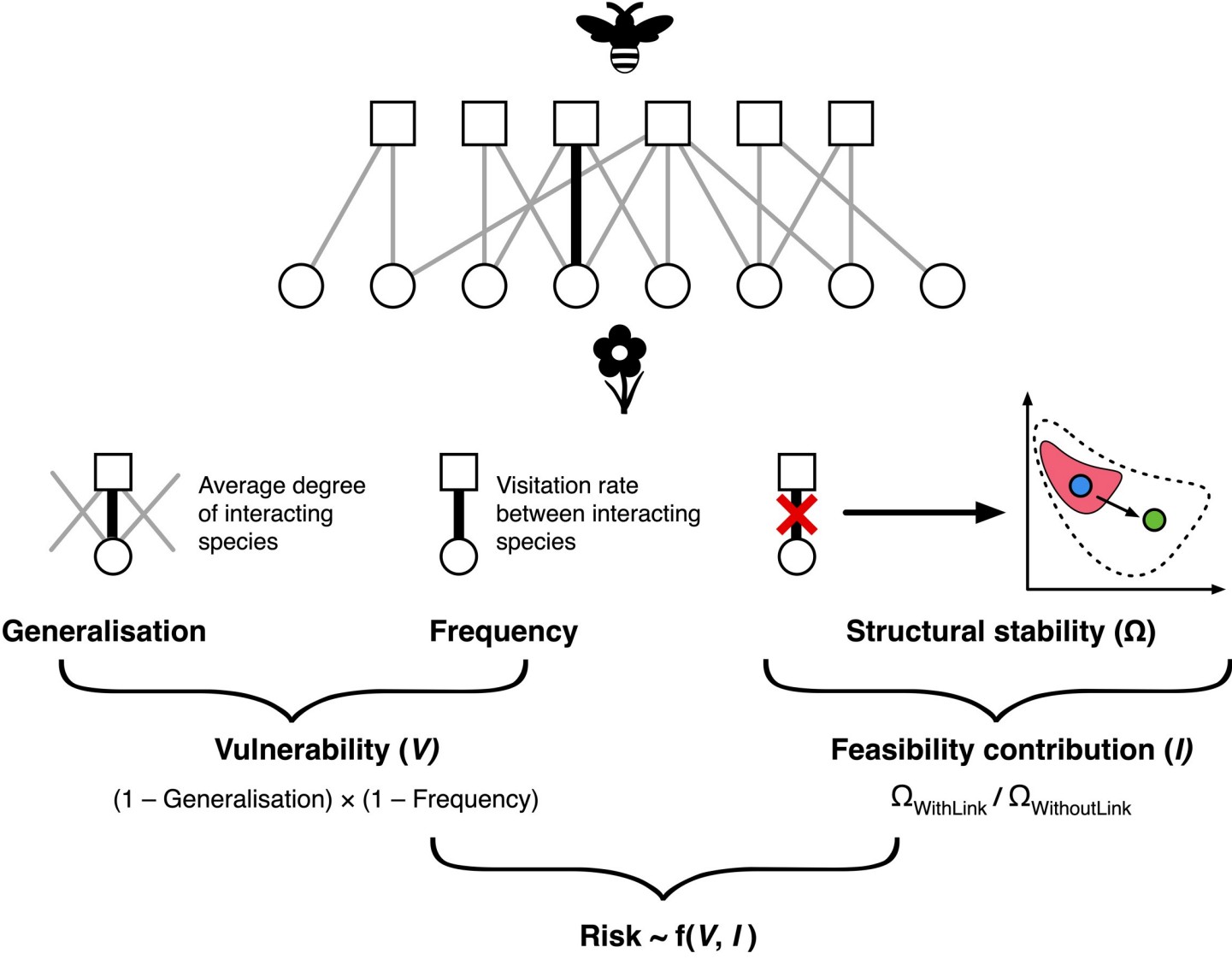

**Fig 1. The quantities used in the analysis.** Top: an illustrative network depicting interactions between plants and pollinators. A focal link is highlighted in black. The generalisation of a link is the average degree of the two interacting species. The frequency of the link is the visitation rate between the two interacting species. Frequency and generalisation are both standardised to between 0 and 1 and then combined to determine the vulnerability of a link, such that vulnerable links are low-frequency interactions between specialists [11]. The feasibility of the network was measured twice: once in its original form, with all links, and once following the removal of a focal link. The ratio of these two values (feasibility with and without the focal link) is the feasibility contribution of the link. This is represented by the graph on the right of the figure. The feasibility of a network is defined as the parameter space of intrinsic growth rates in which all species in a community can have positive abundances [4]. The feasibility domain of the network with the focal link is represented by the dotted outline, and the feasibility domain of the network without the focal link is represented by the pink area. In this case, removing the link has reduced the feasibility of the community (reduced the size of the dotted area). A reduced feasibility means that a perturbation that moves the community from the initial state (blue circle) to a final state (green circle) will result in extinctions because the community moves outside the pink feasibility domain. However, no extinctions would occur under the same perturbation in the original community without the focal link removed, because the final state of the community (green circle) is within the dotted outline. Together, vulnerability and feasibility contribution describe the risk to a community of losing a particular link.

determine, first, whether links that are less vulnerable to extinction are also more important for community feasibility and, second, whether vulnerability and feasibility contribution are intrinsic properties of interactions, given past evidence of the evolutionary conservatism of species roles in ecological networks [30].

## Results

### Relationship between link vulnerability and link feasibility contribution

We calculated the vulnerability and feasibility contribution of 4,330 links from a global quantitative dataset of 29 plant-pollinator and 12 plant-seed disperser networks (Fig 1). Our measure of vulnerability aimed to capture the likelihood of a link between 2 species being lost in the presence of a stressor. An empirical study by Aizen and colleagues [11] examined pollination networks in 12 isolated hills in Argentina that ranged in size from tens to thousands of hectares [11]. They found that interactions in these hills (sierras) tended to be subsets of the interactions present in the next largest sierra. In other words, as the area of habitat decreased, interactions were lost nonrandomly. Two properties of plant-pollinator links independently and additively explained this nonrandom loss: link frequency and generalisation, with less frequent and more specialised links being more vulnerable. We, thus, followed Aizen and colleagues [11] and measured vulnerability as a function of link frequency (how often the 2 species involved in the link interact) and link generalisation (the mean number of links [the mean degree] of the 2 species involved in the link; a form of link 'redundancy'). According to this definition, weak (less frequent) links between specialists are more vulnerable than strong (more frequent) links between generalists [11].

We defined feasibility contribution as the contribution of a given link to the feasibility of a community, where feasibility is a measure of the range of environmental conditions that a community can withstand without leading to species extinctions [4,31]. For our analysis, links with high feasibility contribution were those that, when removed, lowered the community's feasibility; that is, when removed, they reduced the amount of environmental variability that a network can tolerate before species extinctions took place (Fig 1; see Methods). We note that ours is just one possible way to capture the stability contribution of a link. Although other definitions could be used to measure different properties of community stability [32,33], we chose the feasibility approach to avoid constraints on the choice of parameters and type of perturbations that most stability metrics impose [4,34]. Feasibility largely avoids these issues by considering all possible sets of parameters and how these parameters can be changed by perturbations in any direction, thus allowing for a more generalisable and consistent metric of dynamic importance that is comparable across different networks (see Methods).

We found that there was a significantly positive correlation between the vulnerability and feasibility contribution of links across the 41 networks (estimate = 0.11, standard error = 0.015; Wald test: $\chi^2 = 53.71$, df = 1, $P \leq 0.001$; $R^2_{GLMM(m)} = 0.20$, $R^2_{GLMM(c)} = 0.42$) (Fig 2). This correlation indicates that the links that contribute most to the feasibility of our communities have the highest vulnerability values and thus may be those that are most likely to be lost in the face of environmental changes. Frequency and generalisation were also significantly positively correlated, but the relationship was noisy (see S2 Fig), and frequency explained only 8.9% of the variance in generalisation, suggesting that these two components each contribute unique information to the vulnerability metric. Note that our results are not dependent on link asymmetries (the ratio of the degrees of 2 interacting species) because, for a given species, link asymmetry and our measure of link generalisation are perfectly collinear.

### Taxonomic consistency of link vulnerability and link feasibility contribution

What drives the positive relationship between vulnerability and feasibility contribution? Such a pattern could arise if links tend to have the same vulnerability and feasibility contribution independent from the community in which they occur. This would imply some form of

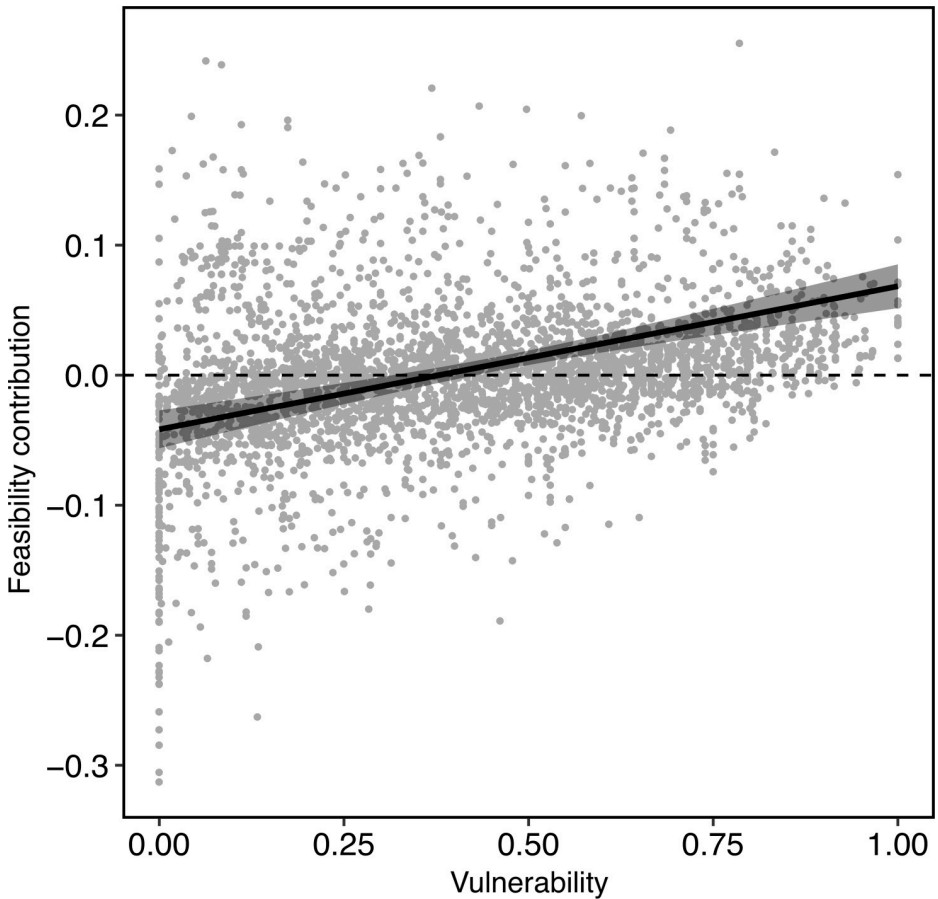

**Fig 2. The relationship between vulnerability (the likelihood of a link being lost) and feasibility contribution (the contribution of a link to a network's feasibility) for all species–species links across 41 mutualistic networks.** Best-fit line is from a mixed-effects model with feasibility contribution as the response variable, vulnerability as a fixed effect, and network identity as a random effect. Grey band represents the 95% confidence interval. To focus on mutualistic effects, these results are from analyses of feasibility contribution with zero interspecific competition ($\rho = 0$), following [17]. See S1 Fig for results using weak competition ($\rho = 0.01$) which were qualitatively similar. Data underlying this figure are given in S1 Data (https://doi.org/10.6084/m9.figshare.12689258.v1).

evolutionary conservatism in interaction properties. We tested this hypothesis by assessing the extent to which vulnerability and feasibility contribution exhibited taxonomic consistency: the tendency for an interaction's vulnerability and feasibility contribution to be more similar across all the networks in which the interaction occurs than expected by chance (see S4 Fig and S5 Fig for an identical analysis of link frequency and generalisation). If vulnerability and feasibility contribution exhibit taxonomic consistency, then all occurrences of a given interaction should have similar levels of these properties. For each interaction, we therefore compared the variance in vulnerability and feasibility contribution to a null expectation where links were sampled randomly from across the dataset (see Methods). If taxonomic consistency was present—that is, if all links in an interaction tend to have more similar vulnerability and feasibility contribution than expected by chance—then the observed variance in vulnerability and feasibility contribution should be less than that produced by the null model. We carried out analyses at genus, family, and order levels, but not at the species level, because very few interactions at the species level occurred more than once in the data. We found a strong tendency towards consistency for vulnerability at all taxonomic levels: between 76% and 83% of interactions had

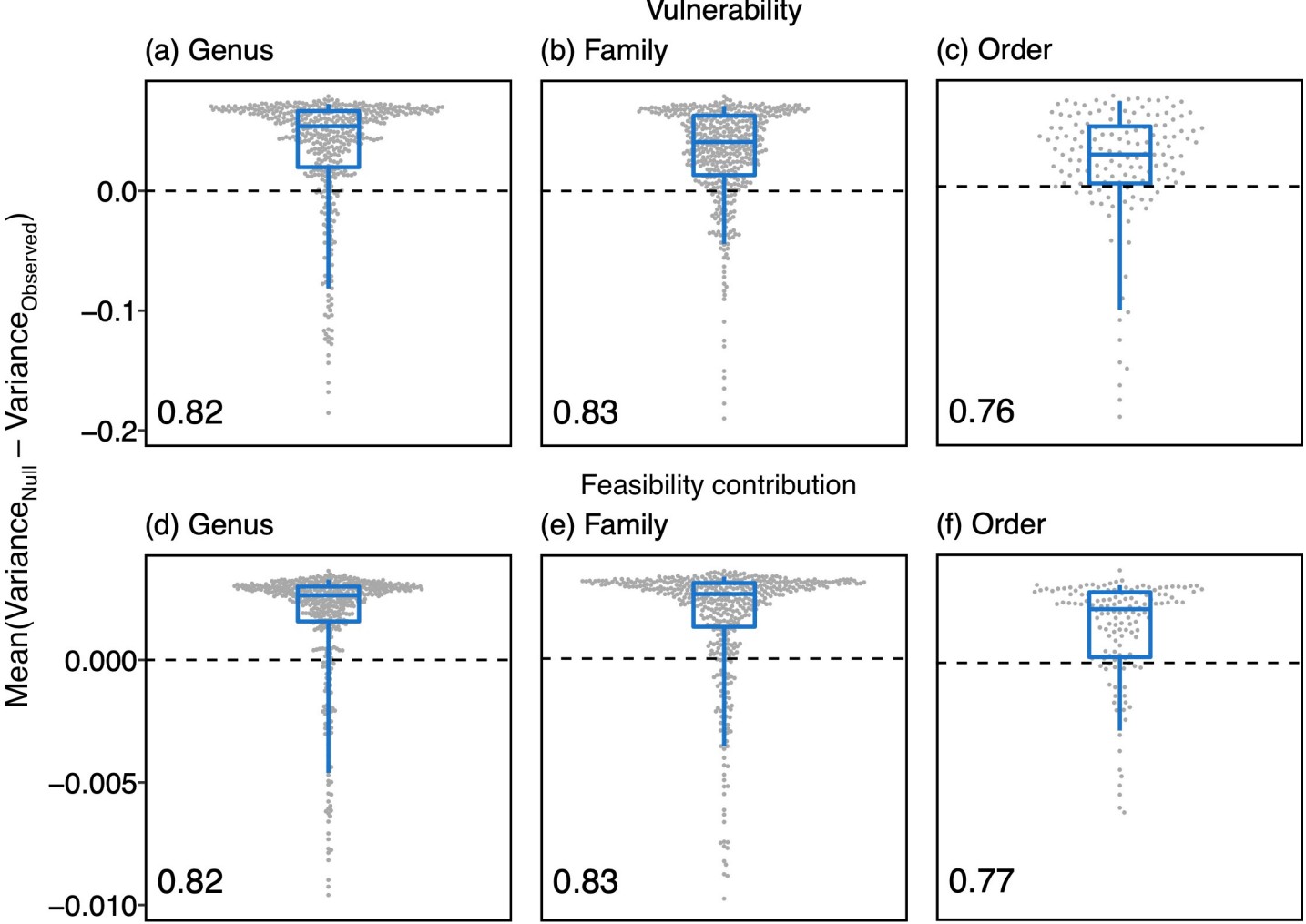

**Fig 3. The degree of taxonomic consistency for each interaction at genus ($n = 469$), family ($n = 466$), and order ($n = 151$) levels, for vulnerability (likelihood of a link being lost) and feasibility contribution (contribution of a link to a network's feasibility).** Taxonomic consistency is the tendency for properties of an interaction to be more similar across occurrences than expected by chance. Points represent individual interactions. Boxplots represent 5%, 25%, 50%, 75%, and 95% quantiles of the same data, moving from the bottom whisker to the top whisker. Number in bottom left of each panel is the proportion of interactions that exhibited positive consistency ($Variance_{Observed} < Variance_{Null}$). For visualisation, a small number of points with low values were removed. The percentage of points with values lower than the y-axis minimum are as follows for each panel: (a) 1.5%, (b) 1.1%, (d) 3.2%, (e) 1.5%, and (f) 1.3%. To focus on mutualistic effects, these results are from analyses of feasibility contribution with zero interspecific competition ($\rho = 0$), following [17]. See S3 Fig for results using weak competition ($\rho = 0.01$) which were qualitatively identical. Data underlying this figure are given in S9 Data (https://doi.org/10.6084/m9.figshare.12689258.v1).

positive consistency in vulnerability (Mean[$Variance_{Observed} - Variance_{Null}$] was positive; proportions shown in Fig 3). There was significant taxonomic consistency in vulnerability for 18% of genus, 17% of family, and 30% of order interactions (see Methods). There was also a strong tendency towards consistency for feasibility contribution at all taxonomic levels: between 77% and 83% of interactions had positive consistency in feasibility contribution (proportions shown in Fig 3). There was significant taxonomic consistency in feasibility contribution for 14% of genus, 20% of family, and 33% of order interactions (see Methods). Given that conservatism was observed across large geographic scales, with many significantly consistent interactions composed of links occurring in different regions or continents, our results suggest that vulnerability and feasibility contribution may be, to some extent, intrinsic properties of interactions and not only a function of ecological context.

## Discussion

To our knowledge, our analysis represents the first attempt to quantify the risk that link loss poses to the dynamics of ecological communities. We find that, across 41 ecological networks, the links that contribute most to a network's ability to tolerate environmental perturbations are the same links that are most likely to be lost in the face of such perturbations (Fig 2). We additionally find that there is a strong tendency for interactions to have more similar vulnerability and feasibility contribution across occurrences than expected by chance, with a substantial proportion of interactions exhibiting this signal significantly (Fig 3).

The positive relationship between vulnerability and feasibility contribution means that the more vulnerable a link is, the more likely it is to have a negative impact on network feasibility if it is lost. This result is not an inherent function of link generalisation. Rather, there are highly generalised links that have a high feasibility contribution and specialised links that have a low feasibility contribution (and even cases in which removing a specialised link increases the feasibility of the network, that is, has negative feasibility contribution) (S2 Fig). Such patterns would not arise if the positive correlation between feasibility contribution and vulnerability was present by definition. Vulnerability is therefore an important indicator of the extent to which a link supports or hinders a community's ability to tolerate environment variation and thus species' long-term persistence. This result is concerning because it suggests that losing vulnerable interactions may reduce the ability of mutualistic networks to absorb future stressors. Aizen and colleagues [11] found low-frequency, specialist links were more vulnerable to habitat loss. Combined with our results, this could mean that environmental stressors like habitat loss may be detrimental for whole-community feasibility, not just those links that are vulnerable.

The positive relationship between link vulnerability and feasibility contribution suggests that links have a tendency to fall into one of 2 categories: low vulnerability and low (negative) feasibility contribution or high vulnerability and high (positive) feasibility contribution. Thus, some links have a high probability of survival to the detriment of the community as a whole, while others contribute to the collective good at the expense of their own viability [35,36]. While the causes of these patterns are unclear, specialised links may contribute more to feasibility because removing links from resource-limited species could strongly condition their existence on their few remaining links, reducing the range of conditions compatible with the stable coexistence of all species in the community. Our results are supported by findings that species that contribute most to a network's nestedness are those that have the lowest survival probability in species persistence simulations [36]. Perhaps there is a tendency for some species to maximise their fitness by being involved in a mixture of 'selfless' links that ensure that the community as a whole remains intact, and 'greedy' links that provide stable benefit to the species over time [35]. Determining how such 'selfless links' arise is an important area for future research because characterising the conditions compatible with their genesis could aid the design or maintenance of resilient ecosystems, and cooperative systems more broadly.

We found that many interactions tend to have similar vulnerability and feasibility contribution values across occurrences, implying a form of evolutionary conservatism in properties of species interactions. While it is hard to decipher the driving force of such observed conservatism, it is reasonable to hypothesise that the consistency of vulnerability is driven by the abundance of the interacting partners. We tested this by assessing consistency in interaction frequency as a proxy for abundance and found strong patterns of consistency (see S4 Fig and S5 Fig). Alternatively, consistency of vulnerability could be driven by conserved patterns of generalisation. For example, pollinator species have been shown to have similar levels of generalisation across their range [37]. Similarly, Rodríguez-Flores and colleagues [38] found that

particular pollinator clades tended to be generalist, while Martín González and colleagues and Rezende and colleagues [39,40] found significant phylogenetic signal in pollinator interactions. Meanwhile, the taxonomic consistency of feasibility contribution has substantial implications. While here we consider the feasibility contribution of individual links, such feasibility contribution values are governed by the whole network structure, not just the roles of the 2 partner species involved in the link. To illustrate this, consider 2 pollinators, $i$ and $j$, and 2 plants, $m$ and $n$, that all interact with each other; that is, $i$ and $j$ both interact with $m$ and $n$. If $j$ leaves the network, the feasibility contribution of all remaining links will change [41]. Thus, that link feasibility contribution is consistent across occurrences suggests that links, and their partner species, are consistently embedded in networks in similar ways, as has been found for species in antagonistic networks [30,41]. While there is strong evidence that plant–animal mutualistic networks continuously change their structure over time as links form and dissolve [42–44], our results suggest that links have similar properties, in terms of vulnerability and feasibility contribution, whenever they do occur. This implies that link consistency imposes constraints on network structure: although networks rewire, such changes in structure result in links having consistent roles [41]. A possible way to test this hypothesis explicitly is to estimate the risk of link loss in communities with temporally resolved interaction networks, as such network data are becoming increasingly available [44,45].

Our results could have useful conservation implications in the future. Differences in links' vulnerability, feasibility contribution, and taxonomic consistency could be used to guide proactive conservation efforts: links with high vulnerability and feasibility contribution could provide a useful starting point to inform prioritisation before any links are lost (see below). Similarly, links with high feasibility contribution that are not currently vulnerable could be a focus of monitoring efforts in case they become vulnerable in the future. Importantly, by explicitly focusing on links themselves, our methods could be used to identify high-priority links that are not expected to be so based only on assessments of species extinction risk. Conversely, our results may be able to inform species conservation if high-priority links tend to involve species that are also of high priority. Determining the relationship between the conservation priority of species and the links that they form is an important area for future research. Our finding of widespread taxonomic consistency potentially allows properties of interactions to be inferred in regions without network data, even if such properties are only known for congeneric, confamiliar, or conorder interactions. This is important because species interaction networks are often cost- and time-intensive to collect, and coverage is highly biased geographically [46,47].

It is important to note, however, that the conservation applications we discuss are, at this time, only potential. Before the methods presented here are used for conservation purposes, it is necessary to empirically validate the models used, as well as their assumptions. We have focused on one aspect of community stability (that of feasibility) and one approach to quantify it. There may be other metrics of stability to consider, and the relationship between vulnerability and stability could depend on the metric of stability used. Similarly, there may be other correlates of mutualistic link vulnerability—perhaps based on species abundance or centrality measures—that would be equal or better candidates for empirical validation. In unipartite food webs, for example, body mass or trophic level could prove useful as a basis for link vulnerability measures [48]. Determining the best proxies of link vulnerability through theoretical and experimental approaches is an important topic for future studies. Our approach nonetheless provides a good theoretical proxy for setting conservation priorities and lays the foundation for future methodological refinement by computational and theoretical ecologists. These refinements, combined with rigorous empirical experimentation by field ecologists, should establish a two-way interaction between theorists and field ecologists that facilitates the development of useful tools for link conservation in the future.

Conserving links is perhaps even more challenging than conserving species. While species conservation requires one species to remain extant, link conservation requires that 2 species remain at sufficiently-high abundance to still significantly interact: 2 species must be prevented from going ecologically extinct [24–26]. Moreover, because interaction extinction often precedes species loss, conservation actions acting on interactions must occur sooner than actions for conserving species. Sampling issues are an additional complication. There is some evidence that interactions are easier to miss than species during sampling [49], suggesting that sampling artefacts could impact interaction conservation more than species conservation (though see [50] for an example where approximately 70% of interactions were sampled). Beyond longer and more intensive surveying, solutions now exist for inferring missing interactions in empirical data, using machine learning [51–53] or statistical methods with network time series [54,55]. These approaches could be important to help overcome the issue of missing interactions in interaction conservation. We found no evidence that sampling effects influenced our vulnerability metric (S1 Analysis).

Despite link conservation attracting little attention so far, the importance of interactions is now widely recognised. Our results could help guide future research in this nascent and important field, because ultimately it is links that support the ecological functions and services that communities provide.

## Methods

### Data

We assembled a dataset of 4,330 plant-animal links from 41 quantitative mutualistic networks spanning a broad geographical range, with data in tropical and non-tropical areas from both islands and mainlands (www.web-of-life.es [56,57]). For the majority of networks, link weights were equal to the total number of visits by animal individuals to a given plant species. The database spanned 2 types of mutualism, comprising 3,182 pollination links from 29 pollination networks and 1,148 seed dispersal links from 12 seed dispersal networks. The data contained 551 plant species and 1,151 animal species. The median sampling intensity [58,59] for the networks was 0.91 (standard deviation of 1.98).

### Interaction properties

**Link vulnerability.** We developed a measure of network link vulnerability following Aizen and colleagues [11]. They identified 2 factors that determine the vulnerability of a mutualistic link between a plant ($i$) and animal ($j$) species: link frequency (hereafter 'frequency') and link degree (hereafter 'generalisation') [11]. Frequency is how often a link occurs between $i$ and $j$ (such as the number of times individuals of a pollinator species visit individuals of particular flower species), whereas generalisation is defined as the mean degree of the 2 species involved in a link, that is, the average number of species with which species $i$ and $j$ interact [11] (see S3 Analysis for a test of the generalisation-vulnerability relationship controlling for species extinctions). This notion of vulnerability aims to capture the sensitivity of a network to the loss of a given link, utilising weighted interaction data (Fig 1).

We calculated the frequency and generalisation of all links in our dataset. Following Aizen and colleagues [11], we first $\log_{10}$ transformed all frequencies. Second, to make results between networks comparable, we standardised frequency and generalisation to between 0 and 1 at the network level. Finally, we calculated the vulnerability of a link between species $i$ and $j$, as $V_{ij} = (1 - f_{ij})(1 - D_{ij})$, where $f_{ij}$ is the standardised link frequency and $D_{ij}$ is the standardised link generalisation. In this formulation, the index can take values between 0 (least vulnerable) and

1 (most vulnerable), which means that it categorizes weak links between specialist partners as more vulnerable than strong links between generalists.

We chose to characterise link vulnerability using the metrics provided by Aizen and colleagues as, to our knowledge, they are the only empirically validated metrics for link vulnerability in mutualistic networks. Additionally, Aizen and colleagues' metrics encompass reasons for link loss beyond species extinction. For example, more specialised links are more vulnerable, not necessarily just because the species involved may be more vulnerable but also because the links cannot be 'saved' by other species if the interacting species become isolated in space or time while remaining extant. Link vulnerability metrics based purely on measures of node vulnerability may not capture this possibility and instead might represent link loss solely as a function of the likelihood of species loss. While this is important, it is not the only aspect of link loss, and thus any future link vulnerability measures must be considered in terms of their ability to capture the full range of reasons for link loss, not just the likelihood of species extinction.

**Link feasibility contribution.** To characterise link feasibility contribution, it was necessary to consider the contribution of a link to dynamic properties of the network. It is common for analyses of network dynamics to use numerical simulations to calculate species persistence: the proportion of species that survive a particular perturbation. Such analyses are useful, but results can be very sensitive to the choice of growth rates: by changing the intrinsic growth rates, it is possible for almost any network structure to have full persistence [4]. Similarly, persistence analyses can be sensitive to the type and direction of the perturbation used; depending on the perturbation specified, a given network structure can appear to have high or low persistence [34]. Thus, simulation approaches do not allow consistent or generalisable conclusions to be reached about the relationship between structure and dynamics in ecological networks.

Feasibility is an alternative, and more general, approach that avoids these issues by not relying on a single set of parameter values or a particular perturbation but instead asking how large the range of parameter values that are compatible with the stable coexistence of all species is [4,31]. Feasibility can therefore be thought of as the 'safe operating space' of ecological communities: it is an indicator of how much environmental stress a community can tolerate before extinction of any of its constituent species. Formally, feasibility is defined as the volume of the parameter space of intrinsic growth rates in which all species in a community can have positive abundances [17,60].

Feasibility is essential for understanding how communities might respond to future environmental changes. For example, in a very feasible community, there is a large range of conditions under which all species stably coexist. Therefore, in the presence of an environmental perturbation, such as climate change or habitat loss, it is less likely that any of the species in the community decline to extinction. Conversely, in a community with low feasibility, there is a small range of conditions under which all species stably coexist. Therefore, perturbations are more likely to result in species extinctions. Regardless of the size of the feasibility domain, there is always complete coexistence of the community within the domain. The size of the feasibility domain simply shows how much change in conditions can be tolerated by the community while all species remain extant.

Calculating feasibility requires a model to describe the population dynamics of species in a mutualistic network. Following several researchers [4,17,31,61], we used a generalized Lotka-Volterra model of the following form:

$$
\begin{cases}
\dfrac{dP_i}{dt} = P_i\left(r_i^{(P)} - \sum_j \alpha_{ij}^{(P)} P_j + \sum_j \gamma_{ij}^{(P)} A_j\right) \\[2ex]
\dfrac{dA_i}{dt} = A_i\left(r_i^{(A)} - \sum_j \alpha_{ij}^{(A)} A_j + \sum_j \gamma_{ij}^{(A)} P_j\right)
\end{cases}
$$

where $P_i$ and $A_i$ give the abundance of plant and animal species $i$, respectively; $r_i$ denotes the intrinsic growth rates; $\alpha_{ij}$ represents intraguild competition; and $\gamma_{ij}$ is the mutualistic benefit. The mutualistic benefit follows the equation $\gamma_{ij} = \gamma_0 L_{ij}/d_i^\delta$, where $L_{ij} = 1$ if there is a link between species $i$ and $j$ and zero if there is no link; $d_i$ is the degree of species $i$; $\delta$ is the mutualistic trade-off [62]; and $\gamma_0$ is the overall level of mutualistic strength. A mean field approximation was used for the intraguild competition parameters, setting $\alpha_{ii}^{(P)}$ and $\alpha_{ii}^{(A)}$ equal to 1 and $\alpha_{ij}^{(P)}$ and $\alpha_{ij}^{(A)}$ equal to $\rho (i \neq j)$. Such an approximation is a standard approach that follows the recommendation of Rohr and colleagues [4] and was necessary given the absence of empirical data on interspecific competition. We estimated the mutualistic trade-off, $\delta$, empirically across all networks in our dataset. $\delta$ is given by the slope of 2 linear regressions [4]

$$\log(f_{ij}/d_i^P d_j^A) = a^P - \delta\log(d_i^P) \text{ and } \log(f_{ij}/d_i^A d_j^P) = a^A - \delta\log(d_i^A),$$

where $f_{ij}$ is the link frequency between animal species $j$ and plant species $i$, $a^P$ is the intercept for plants, and $a^A$ is the intercept for animals. These regressions were performed together on the whole dataset. In this regression, link frequency is used as a proxy for per-capita effects, as this has been demonstrated to be a good surrogate [4,63,64]. We obtained a value for $\delta$ of 0.339, which was consequently used in all simulations (see S4 Analysis for results using a range of $\delta$ values). To focus on mutualistic effects, we ran analyses with zero interspecific competition ($\rho = 0$), following Saavedra and colleagues [17]. Results were qualitatively identical using weak competition ($\rho = 0.01$) (S1 Fig and S3 Fig). The average mutualistic strength was set as half the average mutualistic strength at the stability threshold.

Using this model, the size of a network's feasibility domain is estimated. To do this, no single set of intrinsic growth rates is chosen. Instead, a vector of intrinsic growth rates at the centre of the feasibility domain is first estimated analytically. Being at the centre of the domain, this growth rate vector can tolerate the greatest changes before leaving the feasibility domain. The boundaries of the domain are then approximated by randomly perturbing this central intrinsic growth rate vector to measure the amount of deviation allowed before feasibility is lost (see [4]). In this way it is possible to estimate feasibility: the range of parameter values compatible with complete species coexistence. Using feasibility values, we measured the feasibility contribution of a link between species $i$ and $j$ ($I_{ij}$), defined as the ratio between the feasibility of the network with ($O$) and without ($R$) the focal link: $I_{ij} = \Omega_O / \Omega_R$, where $\Omega$ is the feasibility (Fig 1). Feasibility contribution values were expressed as $([100 * I_{ij}] - 100)$, such that $I_{ij} = 0$ if the feasibility of the network was identical with and without the focal link. Feasibility contribution could not be measured for 931 links which, when removed, resulted in at least one species having no connections; these were excluded from the dataset. The vulnerability of the excluded links was slightly higher than the 3,399 links for which we could analyse feasibility (mean vulnerability of 0.41 versus 0.51 for included and excluded, respectively). From examining histograms of the 2 sets of links (S6 Fig), however, we do not think that this will significantly affect our conclusions because our analysed set still covers the full span of vulnerability values well (over 100 links in the top 5% of link vulnerabilities are from the 'analysed set') and the 2 distributions are similarly shaped (hump-shaped distributions with fewer values at low and high vulnerabilities), just slightly shifted from each other on the vulnerability axis. If we were able to calculate feasibility contribution for all links, and had a complete dataset of 4,300 links, the mean vulnerability would be 0.43, close to the mean vulnerability of the set of links we analysed (0.41).

To examine the relationship between interaction vulnerability and feasibility contribution, we used a linear mixed-effects model, with feasibility contribution as the response variable, vulnerability as a fixed effect, and network identity as a random effect. Linear mixed-effects

models were run and analysed using the 'lme4', 'car', and 'MuMIn' R packages [65–68]. We calculated the variance explained by fixed effects (marginal $R^2_{GLMM(m)}$) and the variance explained by both fixed and random effects (conditional $R^2_{GLMM(c)}$; [37,45,67,69]).

**Link frequency, species abundance, and feasibility contribution.** Our measure of link vulnerability incorporates link frequency which, in turn, may depend on species abundances. We do not believe this is a problem for our analysis, however. The visitation frequencies of a given network in our dataset may be related to species abundances (for which we do not have data). These abundances correspond to a point either inside or outside the feasibility domain. As we do not know the actual abundances of the species in our empirical networks, we can only infer—based on the fact that these communities occur in nature—that the empirical communities are within the feasibility domain. Our analysis shows whether the feasible domain gets bigger or smaller when a link is removed. If it gets bigger, there is a higher probability that the empirical network—with its particular combination of frequencies and (unknown) abundances—is in the feasibility domain, that is, there is a higher probability that the network is feasible without the focal link. Conversely, if the domain gets smaller, it is less likely that the network is in the feasibility domain, that is, there is a smaller probability that the network is feasible without the focal link.

Ideally, our analysis would incorporate a measure of visitation frequency that is independent of, or corrected for, species abundances. However, network data with corresponding species abundances are very rare and perhaps impossible to compile for a global analysis as presented here, with data in mainland, island, tropical, and non-tropical systems.

Thus instead, we briefly review the literature on the topic below, conduct 2 analyses to assess the influence of abundance on our results (S2 Analysis and S5 Analysis), and repeat all analyses with frequency removed as an explanatory variable (S2 Fig and S4 Fig).

Our results suggest that the conventional wisdom that interaction frequency is a proxy for abundance has only mixed empirical support. Although visitation frequency may be influenced by species abundances, it does not represent the true abundance because it is not independent of the network itself [70]. Few studies have looked at the relationship between the distribution of species abundances and visitation frequency. However, those that do exist provide mixed results. Olito and Fox [71] find that abundances do not successfully predict observed interaction frequencies. Hu, Dong, and Sun [72] find that abundance explains only 20%–40% of variation in pairwise interaction frequencies, and Vizentin-Bugoni and colleagues [73] find that 'frequency of interaction is a poor proxy for abundance'. This may be because, while Vizentin-Bugoni and colleagues found a significant relationship between species abundance and frequency of interactions for pollinators, they found no relationship between species abundance and frequency of interactions for plants. Overall, these results suggest a very mixed picture about how closely interaction frequency and abundance are related. These mixed findings are corroborated by our analysis of a plant–hummingbird network that includes independent abundance data [74–76] (S5 Analysis). We found that the relationship between observed interaction frequency and interaction probabilities determined by either empirical or idealised abundance distributions were only very weakly correlated. Thus, we conclude that abundance is a poorer predictor of visitation frequency than traditionally thought, suggesting that our correlation between vulnerability and feasibility might not be biased by the use of visitation frequencies uncorrected for abundances.

## Taxonomic consistency of vulnerability and feasibility contribution

We next assessed the extent to which vulnerability and feasibility contribution exhibited taxonomic consistency: the tendency for an interaction's vulnerability and feasibility contribution

to be more similar across all the networks in which it occurs than expected by chance. Significant taxonomic consistency would imply a form of evolutionary conservatism: that vulnerability and feasibility contribution are intrinsic properties of interactions, rather than a function of the ecological context in which they occur [30].

To explain our analysis, we refer to the example in Fig 4. Fig 4A shows 3 networks. Each of these networks is made up of a number of links, represented by shapes. Network 1 contains 7 links, and Networks 2 and 3 each contain 10 links. The number on each shape is the value of a link property, such as vulnerability or feasibility contribution; in this example, the values are each link's vulnerability. Different shapes (each with their own colour for visual clarity) indicate the identity of an interaction. Thus, the 7 links in Network 1 each belong to one of 5 different interactions.

To assess taxonomic consistency, it is necessary for a link to occur more than once. In the original network data, links were between 2 species. However, only 180 (5.8%) interactions at the species level occurred more than once in the data. To address this, we made comparisons at 3 levels of taxonomic aggregation: genus, family, and order [30]. For each level of taxonomic aggregation, we excluded all interactions that only occurred once. Thus, in this example, we

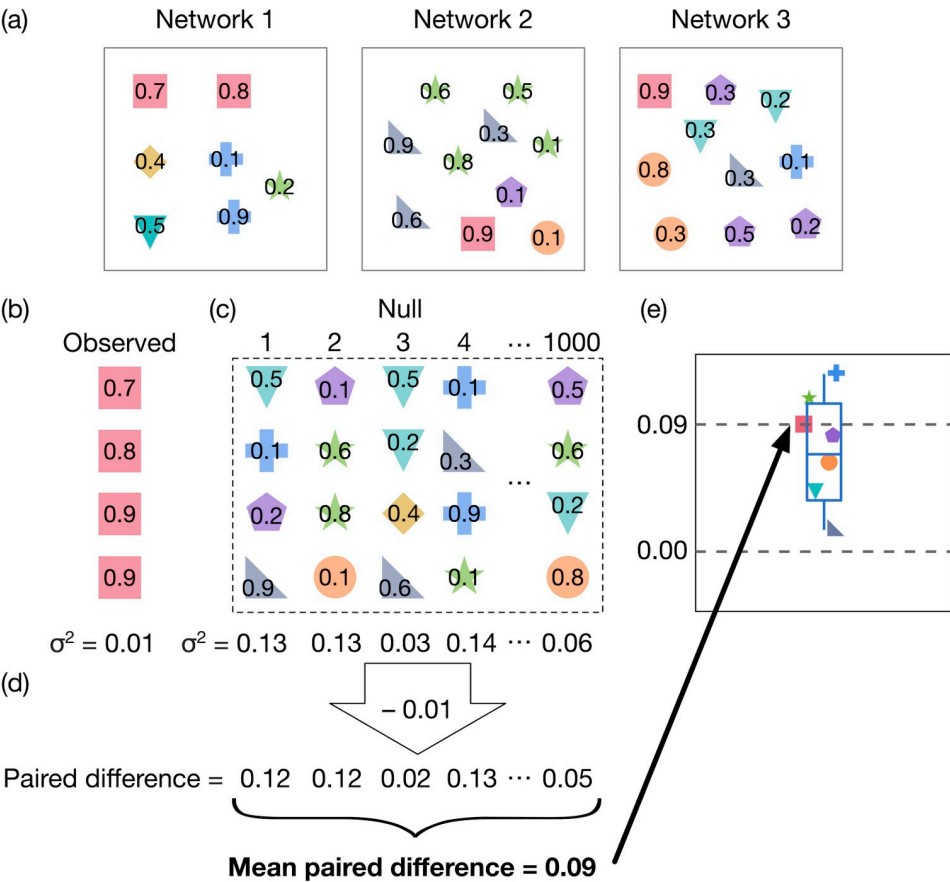

**Fig 4. Schematic illustrating the steps used to assess the taxonomic consistency of vulnerability and feasibility contribution.** (a) Three networks made up of links, represented by shapes. The number on each shape is the value of a link property, such as vulnerability or feasibility contribution. Different shapes (each with different colours for visualisation) represent different interactions. (b) Using the red square interaction as an example, we calculate the variance ($\sigma^2$) in its link property (e.g., vulnerability). (c) 1,000 'null sets' comprising the same number of links as the observed interaction were created, by sampling links randomly without replacement. Variance in each null set was calculated. (d) Paired differences in variance were calculated by subtracting the observed variance from each of the null variances. The mean of these differences was then calculated and plotted (e).

would not assess the taxonomic consistency of the yellow diamond interaction that occurs only once in the data. Note that, due to this taxonomic aggregation, multiple links of the same interaction can occur in a network. For example, in Network 1, one of the red squares could be a link between *Bombus terrestris* and *Knautia arvensis*, while the other red square could be a link between *B. muscorum* and *K. arvensis*. While these links have different species-level identities, if aggregated to the genus level, they are both *Bombus–Knautia* links and thus belong to the same interaction, as shown in the figure. The distribution and number of interactions would be different when links are aggregated to the family and order level.

We then assess taxonomic consistency in vulnerability and feasibility contribution for all interactions, except the yellow diamond as it only occurs once (for clarity, in this example we only consider vulnerability, but this analysis was also carried out for feasibility contribution). We now explain how the analysis proceeds for the red square interaction as an example. If the red square interaction exhibits taxonomic consistency in its vulnerability values, then all red square links should have similar levels of vulnerability. More specifically, variance in vulnerability across all red square links should be lower than expected by chance. We therefore start by extracting all red square links from the data and measuring the variance in their vulnerability, in this case 0.01 (rounded to 2 decimal places for visualisation purposes; Fig 4B). We then assess whether this observed variance is less than a random expectation. To do this, we create 1,000 'null sets', comprising the same number of links as the red square interaction (4) but consisting of links sampled randomly without replacement from across the dataset; links that were part of the focal (red square) interaction were excluded so that they were not sampled as part of any null set (Fig 4C). To ensure that vulnerability and feasibility contribution values were comparable between networks, and to control for any network-level effects, we used only relative values of vulnerability and feasibility contribution. Vulnerability values were rescaled between 0 and 1 at the network level, while feasibility contribution values were already relative (see definition of feasibility contribution earlier). Relative values are more relevant for our study because we were interested in whether interactions tend to have the same relative roles in all communities in which they occur, rather than whether they have the same absolute values of a particular property. For example, we wanted to know whether a given link was always the most vulnerable link in a community, rather than whether it always has an absolute vulnerability value of, for example, 0.7.

For each of these 1,000 null sets, we calculated the variance in vulnerability across all links in the set (Fig 4C). We then calculated paired differences in variance, by subtracting the variance in red square vulnerabilities (0.01) from each of the null variances ($Variance_{Null} - Variance_{Observed}$; Fig 4D). A positive difference between red square variance and a given null variance indicates that the red square links are more similar in vulnerability than the links in that null set. We next calculated the mean of these paired differences (Fig 4D). If an interaction exhibits taxonomic consistency, the mean paired difference will be positive, because the observed variance would be lower than that expected by chance. Finally, we calculated $P$ from the 1,000 paired differences. $P$ is the proportion of null sets that have lower variance in vulnerability than the red square interaction (negative paired differences). An observed interaction had significant consistency when $P < 0.05$, indicating that the probability of a null set having a lower variance than the observed interaction is <0.05. Fig 4E shows how we would plot the taxonomic consistency results (mean paired differences) for all interactions in the example data, as in Fig 3. These steps were repeated for all interactions in the data, for vulnerability and feasibility contribution values (see S4 Fig and S5 Fig for results of this analysis using frequency and generalisation), and for 3 levels of taxonomic aggregation (genus, family, and order). Taxonomic consistency results were qualitatively identical using weak competition ($\rho = 0.01$) (S3 Fig).

'

## Supporting information

**S1 Fig. Relationship between link vulnerability and feasibility contribution ($\rho$ = 0.01).** The relationship between vulnerability (the likelihood of a link being lost) and feasibility contribution (the contribution of a link to a network's feasibility) for all species–species links across 41 mutualistic networks ($\rho$ = 0.01). Best fit line is from a mixed-effects model with feasibility contribution as the response variable, vulnerability as a fixed effect, and network identity as a random effect. Grey band represents the 95% confidence interval. Data underlying this figure are given in S2 Data (https://doi.org/10.6084/m9.figshare.12689258.v1).
(PDF)

**S2 Fig. Relationships between link generalisation, frequency, and feasibility contribution ($\rho$ = 0).** The relationships, for all species–species links across 41 mutualistic networks ($\rho$ = 0), between (a) standardised generalisation and feasibility contribution (Wald test: $\chi^2$ = 50.09, df = 1, $P \leq 0.001$), (b) standardised frequency and feasibility contribution (Wald test: $\chi^2$ = 83.19, df = 1, $P \leq 0.001$), and (c) log(frequency) and log(generalisation) (Wald test: $\chi^2$ = 125.91, df = 1, $P \leq 0.001$). Best fit lines are from mixed-effects models with network identity as a random effect. Grey band represents the 95% confidence interval. Data underlying this figure are given in S3 Data (https://doi.org/10.6084/m9.figshare.12689258.v1).
(PDF)

**S3 Fig. Taxonomic consistency of vulnerability and feasibility contribution ($\rho$ = 0.01).** The degree of taxonomic consistency for each interaction at each taxonomic level, for both vulnerability (likelihood of a link being lost) and feasibility contribution (contribution of a link to a network's feasibility) ($\rho$ = 0.01). Taxonomic consistency is the tendency for properties of an interaction to be more similar across occurrences than expected by chance. Points represent individual interactions. Boxplots represent 5%, 25%, 50%, 75%, and 95% quantiles of the same data, moving from the bottom whisker to the top whisker. Number in bottom left of each panel is the proportion of interactions that exhibited positive consistency (Variance$_{Observed}$ < Variance$_{Null}$). For visualisation, a small number of points with low values were removed. The percentage of points with values lower than the y-axis minimum are as follows for each panel: (a) 1.5%, (b) 1.1%, (d) 7.2%, (e) 6%, and (f) 5.3%. Data underlying this figure are given in S10 Data (https://doi.org/10.6084/m9.figshare.12689258.v1).
(PDF)

**S4 Fig. Taxonomic consistency of frequency and generalisation ($\rho$ = 0).** The degree of taxonomic consistency for each interaction at each taxonomic level, for both standardised frequency and generalisation ($\rho$ = 0). Taxonomic consistency is the tendency for properties of an interaction to be more similar across occurrences than expected by chance. Points represent individual interactions. Boxplots represent 5%, 25%, 50%, 75%, and 95% quantiles of the same data, moving from the bottom whisker to the top whisker. Number in bottom left of each panel is the proportion of interactions that exhibited positive consistency (Variance$_{Observed}$ < Variance$_{Null}$). Considering frequency, there was significant taxonomic consistency for 14% of genus, 19% of family, and 21% of order interactions (see Methods). Considering generalisation, there was significant taxonomic consistency for 21% of genus, 25% of family, and 43% of order interactions. For visualisation, a small number of points with low values were removed. The percentage of points with values lower than the y-axis minimum are as follows for each panel: (a) 1.5%, (b) 0.6%, (c) 0.7%, (d) 11.5%, (e) 8.8%, and (f) 10.5%. Data underlying this figure are given in S9 Data (https://doi.org/10.6084/m9.figshare.12689258.v1).
(PDF)

**S5 Fig. Taxonomic consistency of frequency and generalisation ($\rho$ = 0.01).** The degree of taxonomic consistency for each interaction at each taxonomic level, for both standardised frequency and generalisation ($\rho$ = 0.01). Taxonomic consistency is the tendency for properties of an interaction to be more similar across occurrences than expected by chance. Points represent individual interactions. Boxplots represent 5%, 25%, 50%, 75%, and 95% quantiles of the same data, moving from the bottom whisker to the top whisker. Number in bottom left of each panel is the proportion of interactions that exhibited positive consistency (Variance$_{Observed}$ < Variance$_{Null}$). Considering frequency, there was significant taxonomic consistency for 14% of genus, 19% of family, and 20% of order interactions (see Methods). Considering generalisation, there was significant taxonomic consistency for 21% of genus, 24% of family, and 42% of order interactions. For visualisation, a small number of points with low values were removed. The percentage of points with values lower than the y-axis minimum are as follows for each panel: (a) 1.5%, (b) 0.9%, (c) 0.7%, (d) 11.5%, (e) 8.8%, and (f) 10.6%. Data underlying this figure are given in S10 Data (https://doi.org/10.6084/m9.figshare.12689258.v1).
(PDF)

**S6 Fig. Distribution of vulnerabilities for links for which feasibility contribution could be calculated and links for which feasibility contribution could not be calculated.** The distribution of vulnerabilities for 3,399 links for which we could analyse feasibility contribution ('Included') and 931 links for which we could not analyse feasibility contribution. Vertical dashed lines indicate the mean vulnerability of the 2 groups (0.41 for included links; 0.51 for excluded links). Data underlying this figure are given in S4 Data (https://doi.org/10.6084/m9.figshare.12689258.v1).
(PDF)

**S1 Analysis. Influence of sampling intensity on vulnerability.**
(PDF)

**S2 Analysis. Analysis assessing the influence of species abundance.**
(PDF)

**S3 Analysis. Testing the link generalisation–vulnerability relationship while controlling for local species extinctions.**
(PDF)

**S4 Analysis. Identifying the domain of $\delta$ values where vulnerable links have the highest feasibility contribution.**
(PDF)

**S5 Analysis. Is visitation frequency a good proxy for abundance?**
(PDF)

## Author Contributions

**Conceptualization:** Benno I. Simmons, Lynn V. Dicks, William J. Sutherland, Vasilis Dakos.

**Data curation:** Benno I. Simmons.

**Formal analysis:** Benno I. Simmons, Hannah S. Wauchope, Vasilis Dakos.

**Funding acquisition:** Benno I. Simmons, Lynn V. Dicks, William J. Sutherland.

**Investigation:** Benno I. Simmons.

**Methodology:** Benno I. Simmons, Tatsuya Amano, Lynn V. Dicks, William J. Sutherland, Vasilis Dakos.

**Project administration:** Benno I. Simmons.

**Software:** Benno I. Simmons.

**Visualization:** Benno I. Simmons, Vasilis Dakos.

**Writing – original draft:** Benno I. Simmons.

**Writing – review & editing:** Benno I. Simmons, Hannah S. Wauchope, Tatsuya Amano, Lynn V. Dicks, William J. Sutherland, Vasilis Dakos.

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
