## [Editor Report · Decision Letter 0]

27 Apr 2020

Dear Dr Simmons, 

Thank you for submitting your revised manuscript entitled "Estimating the risk of species interaction loss in mutualistic communities" for consideration as a Research Article by PLOS Biology.

Your revisions have now been evaluated by the PLOS Biology editorial staff, and I am writing to let you know that we would like to send your submission out for re-review.

Please re-submit your manuscript within two working days, i.e. by Apr 29 2020 11:59PM.

Kind regards,

Roli Roberts

Senior Editor

PLOS Biology

---

## [Decision Letter · Decision Letter 1]

2 Jun 2020

Dear Dr Simmons,

Thank you very much for submitting a revised version of your manuscript "Estimating the risk of species interaction loss in mutualistic communities" for consideration as a Research Article at PLOS Biology. This revised version of your manuscript has been evaluated by the PLOS Biology editors, the Academic Editor and the original reviewers.

You’ll see that reviewers #2 and #4 are now satisfied. Reviewer #3 has just one minor request (a reiteration of a previous request) regarding the definition of "importance" in this manuscript; the Academic Editor agrees with the reviewer on this point, saying "[it] indeed seems a questionable term as it usually has other meanings in ecology, so a clearer, more specific term would be more appropriate." Reviewer #1, while being constructive and encouraging, has some more substantial requests which emerge from the last round of revisions; he includes an attachment containing some detailed notes.

In light of the reviews (below), we are pleased to offer you the opportunity to address the remaining points from the reviewers in a revised version that we anticipate should not take you very long. We will then assess your revised manuscript and your response to the reviewers' comments and we may consult the reviewers again.

We expect to receive your revised manuscript within 1 month.

**IMPORTANT - SUBMITTING YOUR REVISION**

*Resubmission Checklist*

*Published Peer Review*

*PLOS Data Policy*

*Blot and Gel Data Policy*

Sincerely,

Roli Roberts

Senior Editor

PLOS Biology

REVIEWERS' COMMENTS:

Reviewer #1:

[identifies himself as Jean-Francois Arnoldi]

[also see attached PDF]

This is my second review of this manuscript by Simmons et al. 

In my first review I had emitted the possibility that the relationship found between link vulnerability and importance was inherent to the definitions of these notions, which discouraged me to dig much deeper in the paper. The authors answered that if there is a connection, it is not a trivial one since it can be strongly weakened by the network complexity (as shown in a supplementary figure showing relationship between link generalism and importance). 

I accept this, but this prompted me to investigate this relationship more thoroughly, in a way that I could understand (see attached document).

My initial critique was basically that vulnerability is defined in a way that makes stronger links more vulnerable, and that strong links are inevitably the most important. But maybe I was wrong. 

I identified that the core ingredient of the paper is the trade-off (delta in the paper) between the generalism of a species (number of interactions) and strength of these interactions. 

Precisely, that this trade-off can lead to situations in which weak links are the most important links (with relates to my previous critique and the author's reply). 

The way one can see that is by imagining removing a link between generalist species. Although this link is weak, because it connects species that have many links, the effects of the removal will ripple trough many species and can thus have a large impact. Note that this reasoning is quite specific to mutualistic networks where one can add up effects ("the friend of my friend is my friend", whereas for negative interactions: "the enemy of my enemy can be my friend" leads to complications when adding up indirect effects). In the document attached I propose that this occurs if overall mutualism (gamma in the paper) is strong enough, while the trade-off delta exists but is not too strong.

My notes also explain why I think that, if the trade-off is strong, it will inevitably be links between specialist species that will be the most important, simply because they are the strongest by construction. I suspect that the authors results hold in the latter case, where the strongest links are the most important. 

Maybe I am wrong in my analysis (it is indeed, approximate and could be improved), but overall I do think that a clear mathematical explanation of the presented results and their domain of validity is missing from Simmons et al's manuscript. 

My notes suggest that such an analysis is not out of reach. 

On the topic of the relationship between link vulnerability and generalism, I came to the opposite conclusion from what I understood the authors suggest. A link is said to be "general" if it connects species that already have many links. In other words, the fact that these species are connected is not surprising, given that they have many links. On the other hand, that two otherwise poorly connected species do interact is unexpected, and thus suggests that their interaction is somewhat special. Which of the two kind of links is the most vulnerable (all else being equal)? At first sight, one would say the latter less expected link, which is what the authors propose, based on a paper by Aizen et al (science 2012). But come to think of it, I find this to be very strange. Indeed, maybe the specialist species themselves are more vulnerable, but not the link between them if the two species are present. In the extreme case, if an insect only feeds on one plant, as long as the insect and plant are present this insect will pollinate that plant. On the other hand, if an insect has many plants to choose from, and some of these plants are visited by many other insect species, it seems plausible that a link between the insect and a highly visited plant would be easily lost (e.g. a slight change in behaviour). I am not an expert of the topic so I won't affirm that I am right. But clearly a strong argumentation is needed here. From what I could get from Aizen et al, link extinction is not separated from species extinctions (and frequency of visit is dot defined per capita, see my point bellow). Again I could be wrong, but there is clearly something a bit subtle here that requires some explanation. 

The irony is that, by disagreeing twice with the authors, I end up being in agreement with their overall claim: that the most vulnerable species can be the most important. But only if mutualism is large enough and the afore mentioned trade-off is not too strong (I think that, in the authors current view, it would be the opposite conditions that would allow the author's claim to hold). 

I recommend that the authors delineate more clearly the domain of validity of their claim, and justify more thoroughly the many assumptions that go into their analysis. In particular, I find the statistical analysis leading to a value of delta very arbitrary. Furthermore, it is based on records of the frequency of insects visits (fij in the text). Maybe I missed something, but those frequencies are likely to be strongly dependent on insect and plant abundance and thus should not directly inform per-capita interactions (which is what the paper is about, at least the method to define link importance). If I am wrong and fij really relates to per-capita interactions, why not use it directly to parametrize link strength? delta would then be a phenomenological parameter used to explain the results (in terms of trade offs as explained above and in my notes). At this point I suggest to treat delta as a free parameter and identify the domains in which vulnerable links are the most important. Overall this paper should aim at proposing clear hypothesis and not make definite ecological claims, as there are too many parts of the analysis that rely on arbitrary choice of parameters (strength of mutualism, strength of completion, trade off etc...). 

Although I am critical, I must say that the paper is very well written and (obviously) thought-provoking. The idea of looking at the effects of perturbing interactions (what we called structural stability in paper with Bart Haegeman from 2016) is something that I personally think is very important and generally overlooked, and I am glad that the authors thought to investigate it. I sincerely hope that this review and attached notes will be helpful. 

Best

Jean-Francois

Reviewer #2:

[identifies himself as Shawn J. Leroux]

 The authors have done an outstanding job of addressing the many reviewer comments. The additional details in the methods (including some new analyses) and revised introduction and discussion have greatly improved the ms. I have no additional concerns.

Shawn J. Leroux

Reviewer #3:

I appreciate the effort that the authors have put into answering the concerns I raised in my previous review. The manuscript is very well written and there is merit in what the authors present in this manuscript. Moreover, I was pleased to see the changes made to the manuscript, including new figures, clarifications and additional discussion points provided by the authors.

While some of my concerns regarding how much the authors' "importance" actually reflects ecological importance remain, I understand that using simulations to understand the identity of affected species due to the vulnerable link removals might not be a trivial exercise. My concerns are perhaps little more than a matter of terminology. Namely, I am still troubled by the use of the word "Importance" to define the 'contribution to' a well known network property. I think it's a bit misleading, and I *personally* feel that something along the lines "contribution to feasibility" would be much more appropriate. That said, I appreciated the clarifications added by the authors across the manuscript regarding this point.

Reviewer #4:

[identifies himself as Ignasi Bartomeus]

Thanks for the super detailed response and the extra analysis, which makes the manuscript very robust. I don't have any further comments and I congratulate the authors on this nice study.

---

## [Decision Letter · Decision Letter 2]

9 Jul 2020

Dear Dr Simmons,

Thank you for submitting your revised Research Article entitled "Estimating the risk of species interaction loss in mutualistic communities" for publication in PLOS Biology. I've now obtained advice from one of the original reviewers and have discussed their comments with the Academic Editor. 

Based on the remaining review, we will probably accept this manuscript for publication, assuming that you will modify the manuscript to address the remaining points raised by the reviewer. Note that while the reviewer marks some of his points as "mere suggestions," we urge you to address them as fully as you can. Please also make sure to address the data and other policy-related requests noted at the end of this email.

We expect to receive your revised manuscript within two weeks. Your revisions should address the specific points made by each reviewer. In addition to the remaining revisions and before we will be able to formally accept your manuscript and consider it "in press", we also need to ensure that your article conforms to our guidelines. A member of our team will be in touch shortly with a set of requests. As we can't proceed until these requirements are met, your swift response will help prevent delays to publication.

*Copyediting*

*Published Peer Review History*

*Early Version*

*Submitting Your Revision*

Sincerely,

Roli Roberts

Senior Editor

PLOS Biology

DATA POLICY:

Many thanks for providing the links to the raw data. However, we also need the individual numerical values that underlie the data summarized in the figures and results of your paper be made available in one of the following forms:

Regardless of the method selected, please ensure that you provide the individual numerical values that underlie the summary data displayed in the following figure panels as they are essential for readers to assess your analysis and to reproduce it: Figs 2, 3ABCDEF, S1, S2ABC, S3ABCDEF, S4ABCDEF, S5ABCDEF, S6, S7, S8, S9. NOTE: the numerical data provided should include all replicates AND the way in which the plotted mean and errors were derived (it should not present only the mean/average values).

REVIEWER'S COMMENTS:

Reviewer #1:

[identifies himself as Jean-Francois Arnoldi]

The authors did a great job at addressing my previous comments. In particular, the analysis of the cited paper by Aizen et al (science 2012) is quite helpful. 

Furthermore, the new figure where delta is varied is indeed the kind of results that I though was missing in the previous version. Note that it could be useful to see the impact of gamma (strength of mutualism, here it is defined so that the dominant eigenvalue of the interaction matrix is -0.5, but one could look at cases where this eigenvalue is -0.9, weak mutualism, and -0.1, strong mutualism, for instance). 

My remaining comments are the following, since other reviewers were already satisfied with the previous version I provide those comments as mere suggestions:

In the new figure where delta is varied, I don't think that it makes sense to compute vulnerability, since it requires the frequency of visits (fij fixed from observations) which in turn ought to determine delta (which is now varied). Thus I would suggest to plot the slope of specialism in contribution to feasibility against delta. I am also surprised that the slope is positive and strongest when delta is zero (no generalist-strength tradeoff), as I would naively expect the opposite. I encourage the authors to double check their simulations and see if they can provide an explanation. It would be worthwhile to see if the results are specific to the empirical networks or would also hold for a random bipartite network with comparable variance in number of links per nodes. 

Finally, I am still not convinced by the use of frequency of visits in the analysis. Indeed, maybe it can be used to infer the tradeoff delta (although I don't really get it), but it is clear that this frequency depends on abundances. This is not consistent with the feasibility analysis that starts from a growth rate vector where all species have the same abundances and then looks for the size of perturbations of that growth rate vector that leads to extinctions (and thus varies abundances widely, so frequency of visits can not be fixed). I thus encourage the authors to provide results on specialism vs contribution to feasibility instead of the empirically constructed measure of vulnerability of Aizen et al.

---

## [Editor Report · Decision Letter 3]

31 Jul 2020

Dear Dr Simmons,

On behalf of my colleagues and the Academic Editor, Michel Loreau, I am pleased to inform you that we will be delighted to publish your Research Article in PLOS Biology. 

Early Version

PRESS 

Kind regards,

Vita Usova

Publication Assistant, 

PLOS Biology

on behalf of

Roland Roberts,

Senior Editor

PLOS Biology